# Assessing and Responding to Suicide Risk in Children and Young People: Understanding Views and Experiences of Helpline Staff

**DOI:** 10.3390/ijerph191710887

**Published:** 2022-09-01

**Authors:** A. Jess Williams, Juliane A. Kloess, Chloe Gill, Maria Michail

**Affiliations:** 1Institute for Mental Health, School of Psychology, University of Birmingham, Birmingham B15 2TT, UK; 2Institute of Mental Health, School of Medicine, University of Nottingham, Nottingham NG7 2TU, UK; 3Department of Informatics, School of Natural, Mathematical & Engineering Sciences, King’s College London, London WC2R 2LS, UK; 4School of Psychology, University of Birmingham, Birmingham B15 2TT, UK; 5Research and Evidence Department, National Society for the Prevention of Cruelty to Children, London NW1 0AP, UK

**Keywords:** suicide, helplines, counselling, thematic analysis, interviews

## Abstract

Introduction: Suicide is a key issue impacting children and young people. Helplines offer unique benefits, such as anonymity, varied communication avenues and low cost, which help to promote help-seeking behaviour. The aim of this study was to explore the views and experiences of helpline organisations of identifying, assessing, and managing suicide risk among children and young people. Methods: Thirteen professionals from three UK-based helplines and online counselling services took part in semi-structured interviews between November 2020 and January 2021 via Zoom. Interviews were transcribed verbatim and analysed using Thematic Analysis. Results: Three superordinate themes were identified: (i) Starting conversations about suicide; (ii) Identifying and responding to “imminent” suicide risk; and (iii) Responses to suicide risk in relation to safeguarding. Limitations: Recruitment was limited by COVID-19 due to the demands needed from helplines at this time. Conclusions: Our findings highlight not only the different types and range of services helpline organisations offer to young people who might be at risk of suicide, but most importantly the distinct role they have in young people’s help-seeking pathway.

## 1. Introduction

Suicide is the fourth leading cause of death for those aged between 15 and 19 years [1]. In 2020, there were 4.9 suicides per 100,000 for this age group within the UK [2]. While this was a decrease from previous years, it has been suggested that this decrease may be related to a delay in death registrations during the COVID-19 pandemic [2]. Furthermore, UK-based evidence suggests that during the pandemic, there was an increase in suicidal thoughts and self-harm [3]. Self-harm is a significant risk factor for suicide [4] and among 10–18-year-olds, suicide rates are over 30 times greater than the general population in the year following self-harm behaviour [5].

Young people with suicidal experiences often do not access mental health services [6,7] or are unwilling to seek professional help [8]. Previous research has identified a number of factors that help explain this, namely, being unsure of where or who to seek help from [9], fears of stigma from staff [10,11], poor communication surrounding suicide from professionals [12], and concerns around confidentiality and disclosing suicidal thoughts or behaviours [13,14]. Disclosure may be a significant challenge among children and adolescents, due to fears that parents or guardians will be informed. For example, in some areas of the U.S., parents have legal access to adolescents’ electronic health records until they are 18 years old or are notified when services are sought [15], thereby nullifying the individual’s confidentiality when seeking professional help. 

When young people do try to access mental health services, there are often barriers. In the UK, these include short GP consultations, leaving insufficient time to discuss mental -health-related concerns, long waiting lists for treatments [16], and poor communication between services hindering referral pathways [17]. During the pandemic, these barriers were exacerbated, partly due to a lack of resources and capacity to respond to the unprecedent demand for mental health support among young people and partly because of services having to adapt their working practices as a result of COVID-19 [18,19]. Many young people were also concerned about burdening the NHS at a time of increased demand [19]. Thus, alternative avenues for support need to be explored. 

Young people are more willing and likely to access support from community-based services (e.g., online forums, peer support groups, helplines) when feeling suicidal [8,14]. From a previous meta-analysis, when young people did seek help, it was more often from community-based supports (40–68%) rather than medical or mental health services, which were classed as professional help-seeking (below 50%) [8]. Helplines are well-placed to provide young people with support for suicidal thoughts and feelings through various communication pathways (telephone-based, text-based, webchat or email supports). The benefits highlighted by young people suggest the ability to be anonymous, and sense of control over the conversation as important when seeking help from a helpline [20,21]. Furthermore, helplines are often highly accessible when compared to face-to-face services, with options for making contact. Helplines thereby offer an alternative point of access for support [22], bypassing fears around help-seeking from professional services [10,11,12,13,14].

Despite being well-placed for user access, helplines face their own challenges for staff and volunteers. For example, helpline staff need to determine the level of emotional need from callers [23] which requires a level of rapid assessment of genuine dis-tress while also dealing with potentially abusive or inappropriate callers [23]. Furthermore, volunteers often need to make judgements of who is in need of assistance which is directly against the helpline ethos of being non-judgment [23], this may then damage supportive relationships with callers. It seems that the limited face-to-face dynamic of helplines, although helpful for preserving anonymity, could create barriers for staff or volunteers to gain a clear representation of distress and callers feeling as though they are being dis-missed [23]. During COVID-19 pandemic, additional demand impacted helplines. Between 23 March and 10 May 2020, the NSPCC Childline service delivered 30,868 counselling sessions to children and young people, 36% of which were about mental and emotional health concerns, 13% were about suicidal thoughts and 6% about self-harm [24]. Samaritans also saw a 12% increase in calls made between 2 a.m. and 6 a.m. compared with previous years and longer phone sessions (40%) [25]. This increased demand for helpline services means that staff and volunteers are under greater strain to aid a higher number of callers while still making precise and calculated evaluation of caller wellbeing.

Helplines are a common source of help-seeking among suicidal young people. Given the recent heightened demand and challenges for helpline services, it is important to understand how helpline staff and volunteers identify, assess, and manage suicide risk among young people, as well as the challenges staff face when responding to suicidal young people, and how they manage these within an organisational structure. Having a better understanding of the current practice and challenges facing these services could help us identify examples of good practice and examples where practice could be optimised, thus highlighting opportunities for improvement through, for example, the provision of tailored training and/or resources. The aim of this study was to explore the views and experiences of helpline staff and volunteers when identifying, assessing, and managing suicide risk among young people in order to better understand their current practice. 

## 2. Materials and Methods

### 2.1. Context

The study took place during the COVID-19 pandemic. Helpline professionals report-ed the increasing demand of children and young people seeking help from their organisations during this period. Furthermore, the distress reported by young people appears to be on the rise, with more complex needs being discussed [26]. In addition, services struggled with staff absences, and changing working practice. This is in line with challenges experienced across mental health services in the UK during this time. Due to this increased demand of both complex needs and general help-seeking, it seems timely to have a deeper insight into how helplines deal with young people seeking help for suicide. Due to the pandemic, all methodology took place online rather than in-person settings.

### 2.2. Participants

Thirteen professionals from three UK-based helplines and online counselling services took part in the study between November 2020 and January 2021, using snowball sampling. The eligibility criteria for participants were: (i) to currently be working at a helpline or online counselling service, (ii) to have experience dealing with service users with suicidal experiences, and (iii) to be able to take part in an English-speaking online interview. Participants were members of staff from a student helpline (*n* = 5), an online counselling and support network (*n* = 5) and a self-harm specific helpline (*n* = 3) and represented different levels of seniority and roles from within their organisations (e.g., listening volunteers, online counsellors, suicide trainers, welfare or safeguarding team members, senior practitioners, or head of service). No participants withdrew from the study. 

Two of the services were traditional helplines with phone, text, webchat and email communication streams. The online counselling and support network hosted a helpline chat function alongside scheduled online counselling appointments and peer networking forums. Given the different language used by services, we will be adopting the term “service users” (SU) when referring to those accessing the service in any manner, and “frontline staff” when referring to staff or volunteers who respond to SU contacts. 

### 2.3. Procedure 

Ethical approval was granted by the lead author’s primary university. Recruitment emails were circulated to mental health helplines including those dealing with service users (SUs) at risk of self-harm and suicide specifically. 

If a service agreed to participate, they were asked to send an internal email to staff members informing them of the study. This invited staff to contact the lead author, the team research assistant, to express their interest in taking part. Interested staff were subsequently provided potential participants with an information sheet, a consent form, and further information about the interview process. Once a signed consent form had been received by the research team, an interview was set up via Zoom at a date and time convenient to the participant. Interviews lasted between 22 and 81 min (*M =* 56 min). 

Prior to each interview, the researcher talked participants through the practicalities of Zoom interviews (e.g., what to do if the connection was lost), explained the study again in detail, reminded them of their rights to withdraw, stop or pause the interview at any time, as well as giving them an opportunity to ask questions. Following the interviews, all participants were thanked for their time and invited to ask any questions. 

### 2.4. Data Collection

The research team developed a semi-structured interview guide which aimed to explore (i) helpline staff’s experiences of identifying, assessing and managing suicide risk in SUs, (ii) provision of suicide training and supervision within the service, and (iii) barriers to and facilitators of implementing protocols and policies at the service. This was available on request to participants prior to the interview. 

All participants were interviewed solely by the first author, who has extensive experience of online interviewing with suicidal youth and clinical professionals. Interviews were conducted online where-ever participants felt comfortable that they could speak openly about their experiences, typically this was at work or at home. Field notes were taken during the interviews, which acted both as question prompts and highlighted key points in relation to the research questions. All interviews were audio-recorded using a Dictaphone and transcribed verbatim by a professional transcription service. Any personally identifying information was removed.

### 2.5. Analysis

The study is reported in line with the Consolidated Criteria for Reporting Qualitative Research ([27], see Appendix A). The transcribed interviews were analysed using Thematic Analysis, and the analytical process followed the recommended six phases by Braun and Clarke [28,29]. Following transcription, all transcripts were imported into NVivo 12, and the first author began coding these inductively. Extensive coding of the transcripts was performed, generating over 4000 individual codes. Codes were clustered to develop a meaningful preliminary framework in line with the research questions, which were formulated prior to engaging with the data. The codes and preliminary framework were then reviewed with the second author. Subsequently, themes were developed and discussed within the research team, with any additional considerations and reflections being incorporated. All members of the research team reviewed the final framework.

## 3. Results 

Participants shared their views and experiences of working with SUs at risk of suicide within helplines or online counselling services. From this, three superordinate themes were identified: (i) Starting conversations about suicide; (ii) Identifying and responding to “imminent” suicide risk; and (iii) Responses to suicide risk in relation to safeguarding. Each theme is presented in more detail below and supported by participant quotes. An overview of the themes is presented in Table 1.

### 3.1. Starting Conversations about Suicide

This theme highlighted the importance of frontline staff starting a conversation about suicide. It was clear that frontline staff aimed to create a safe space in which children and young people felt that they could openly discuss their suicidal thoughts, feelings or experiences. This was achieved through explicitly bringing the topic of suicide into a call or chat, and by aiming to explore the context or situation around suicide risk. 

#### 3.1.1. Explicitly Asking about Suicide

Across services, there were several ways in which helpline staff described directly initiated suicide-related conversations with SUs. Often, it appears that this was facilitated by explicitly asking them if they were struggling with thoughts about self-harm or suicide: 

*“…then we would actually ask that question in quite an overt way, saying, “It sounds like you’re really struggling–can I ask, are you having thoughts or feelings…it sounds like you’re having, you know, thoughts of suicide…” and usually that tends to provoke a response of “No, that’s not what I meant” or “Yes, actually, I am” and then that discussion [relating to suicide] opens up.”* (S1P1).

By introducing the topic of suicide, helpline staff described offering SUs a space to respond openly regarding their thoughts or feelings of suicide. This line of questioning could either move the conversation away from suicide, if it was shown as not relevant to that individual or opened up a space for them to discuss it if they wished or needed to. 

*“You know, we explicitly ask, you know, on contacts around that because sometimes I think, if it’s not explicitly asked, you don’t give a person a space to say yes or no. And of course that changes, you know. It can be from fleeting thoughts to really intense, overwhelming thoughts.”* (S3P2).

Some participants’ accounts indicated that they felt that SUs may not have been asked about suicide in other services so overtly, such as by their GP, and it was therefore necessary to demonstrate that suicide is a topic which can be talked about openly and candidly in their service. 

#### 3.1.2. Exploring and Understanding

From participant accounts, once a SU felt comfortable to discuss their experiences around suicidal thoughts, feelings or behaviours, helpline staff stated that they spent time exploring and understanding their situation and the context within which the SU found themselves. This could be through asking about how the SU had been feeling that day or what might have brought forth their suicidal thoughts or feelings. 

*“…like “How long have you been feeling like this. You know, like has something like led to this tonight or how you’re feeling today?” and then like, if they give us like a feeling, we kind of ask questions down like, well, “Why are you feeling like this?” and kind of try and delve deeper into like their story and keep them talking about that…”.* (S2P3).

Some frontline staff described that they would extend this line of questioning and active listening to get a better insight into what was going on in the SU’s life. This would often include considering whether suicide is something that the individual had been struggling with on a longer-term basis, what tended to trigger their thoughts, feelings or behaviour, and whether there was someone to support them when they were feeling this way. 

*“I would talk about, obviously, historical, because that may come up–you know, is it actually the first time? You know, how long have they been feeling like this, you know, and have they had a historical, em, time of, you know, trying to commit suicide, em, because, again, I think that’s really important to…to get a full picture of actually what their life and journey has been like...”* (S3P1).

*“I kind of always start with like, “I’m really sorry you’re feeling like that–tell me a bit more, like has something happened recently that’s making you feel like you need to do this?” or trying to get to what’s kind of…what’s the cause of it, really, or if the young person is aware of any causes, because, quite often, they’re not. They’re just like, “Well, I’ve got these feelings, I don’t know why, I don’t know where it’s come from,” and that sort of therapeutic work is about working out with them something that may have triggered it, and [I’m] thinking…finding a trigger helps the young person understand it more…”.* (S3P3).

By exploring suicide with the SU, participants described how this helped them to improve their understanding of their own emotions and thoughts, as well as articulating the experiences more clearly for the staff member. This facilitated a shared impression of what the SU was struggling with and how to precede. 

### 3.2. Identifying and Responding to “Imminent” Suicide Risk

Across the services, there was no firm consensus of what qualified “imminent” or “immediate” risk; therefore, participants described their own definitions. This was typically a suicide plan within a specific timeframe. Identifying and responding to risk was clearly related to the varying approaches to suicide risk and autonomy taken by the ser-vices, e.g., reactive safeguarding. It was obvious that there was not one single way universe-sally to identify or respond to “imminent” risk. 

#### 3.2.1. Determining Immediacy of Risk 

There was variation between participants in how they determined “immediate” or “imminent” risk for a SU. This was linked to the helpline’s threshold for safeguarding. For example, some services would not consider self-harm as an immediate risk of suicide. One helpline labelled self-harm as “passive suicide” rather than “active”, which was more closely associated with suicidal intention. In other services, determining the degree of immediacy of risk was linked to how SUs described their self-harming behaviour, including the severity of the injury: *“…em, but it is that sense of, em, you know, if someone, for example, for cutting and they were describing it in such a way that felt that they were in immediate danger, then, em, yeah, that would be what would kind of instigate…”* (S1P2).

Another mechanism for determining “imminent suicide risk” focused on understanding the SU’s intention behind their thoughts and feelings, including if there was a suicide plan and the timeliness of this plan.

*“I may say, actually, like, “Is there intention of you wanting to take your life?” and em…yeah, I guess it depends where they’ve gone from there. I just…I’m really trying to think about kind of important questions I would actually ask in that situation… Sometimes, I will say like a scale, em, because people may say, “I’m not sure” or, em, and kind of just saying like, “Out of 1 to 10, you know, if 10 is you’re definitely going to do it, where do you feel you’re at right now?” and I think it’s really important to say “right now” because, em, obviously that could change all the time.”* (S3P1).

Overall, participants presented “imminent” risk as being associated with a suicide plan with a specific timeframe. This became more evident if the SU was actively engaging with that plan: *“If they’ve actually taken stuff that night, then obviously it’s more urgent…”* (S3P3).

#### 3.2.2. Making Judgement Calls

Participants discussed how they often had to make their own judgements as to whether a call or chat would trigger safeguarding procedures. This requires a high level of trust from the helpline in their staff members that they have the skills and are confident to make these decisions: *“…I think we obviously have to trust volunteers to make their own judgement. Sometimes, there will be a certain amount of kind of a sensing of how risky it feels…”* (S1P2). 

One participant discussed how helpline staff responding to the chats, calls, texts or emails were in the best place to make assessments of “imminent suicide risk” as they are the ones holding the conversations with the SU: *“…because you were in that relationship with that person offering the frontline support, it is, you know, we will trust your judgement on whether we need to make that [SAFEGUARDING] call or not…”* (S1P1).

Alongside the need to make a judgement call of immediacy, some participants also discussed how they did not want to rush into making a decision which could trigger safeguarding. This was predominantly related to the risk of jeopardising the therapeutic relationship with the SU. Frontline staff related that they would usually weigh up the risks and benefits, allowing the SU space to explore their suicidal thoughts and feelings, and make a decision relating to safeguarding when the time was right, then proceeding accordingly. 

*“It’s something I would definitely negotiate. It isn’t something that–when I say “negotiate”, if there is immediate risk, I would act on it, but I wouldn’t just say… It all depends on the situation [laughing]. If someone gave details, it’s not like, right, I have to do this right now! I would be certainly listening to…to…. I think it’s that, giving them that space to explore it, and if, at the end, it does need to go to the Police, then it would.”* (S3P1).

### 3.3. Responses to Suicide Risk in Relation to Safeguarding

Services had similar procedures in place when responding to a SU who was at risk of suicide. However, participants critically reflected on the utility of safeguarding procedures. Safeguarding responses are also reliant on the individual offering personal details. There were therefore cases where helpline staff were unable to respond to individuals presenting with suicide risk due to a lack of information. 

#### 3.3.1. Triggering Safeguarding 

All services required a basic set of information, including name and location, to trigger a safeguarding response when a SU presented with imminent suicide risk. These details could be offered voluntarily by the individual, consenting to their use in the event of helpline staff being concerned for their safety. However, two services also recorded personal details shared by SUs as a safeguarding precaution, which was without explicit consent from the SU. These details could therefore also be used in instances where the SU presented as a risk to themselves, or where the individual specifically requested to trigger a safeguarding response. Safeguarding responses require frontline staff to break the service’s confidentiality policy: *“So, em, yeah, then, basically, if they’ve asked for the ambulance, then we get their details, and that’s when we break our confidential policy then because, obviously, we don’t normally do that…”* (S2P1).

*“If–there are some callers who have disclosed their information, so…somebody potentially could disclose that during a call, so saying where they are or, you know, giving the location, and that then would be around making a decision about if that person was at immediate risk, and if they are, then we would let them know that we were going to get help for them, and we’d end the call and call the Police so…or call 999…”*. (S1P1).

The standard safeguarding process across all services was to call emergency services. Participants highlighted that it was important to be clear about their intentions regarding (i) storing personal details, (ii) confidentiality and its limits, (iii) when safeguarding might be necessary, and (iv) who was being called. This was often communicated when SUs started to present with suicidal distress. Participants discussed the importance of maintaining the relationship with the SU, which was often benefitted by the frontline staff member being able to judge the context and situation themselves, rather than follow a closed script. Staff felt that this enabled SUs to have a greater sense of trust in the discussion, enhancing rapport and satisfaction with the contact, as well as allowing for open disclosure of suicide risk. According to the staff, this flexibility allowed them to solidify relationships and be transparent when they felt there was a need to break confidentiality:

*“…being really transparent with the young people, letting them know this is what we’re going to do, because I really care and I’m worried about you, and following up and making sure that they know that we’re still there. So, they still get supported through all that, which is great, and then obviously the follow-up, em, if…in…circumstances where we have contacted the Police to complete a welfare check because we were concerned or an ambulance has been called.”* (S3P5).

Despite clear communication and policies regarding safeguarding procedures, par-ticipants were concerned about the impact of calling emergency services for SUs, and con-tributing to their distress further. This was a challenge for some participants as they weighed the judgment of risk compared to further harm to the SU. Sometimes SUs would request that police were not called: *“There is often a kind of “Please don’t contact the Police”, especially with people that we already have contact details for, and that’s a really difficult conversation…”* (S1P3). This made for uncomfortable situations as the frontline staff needed to ensure safeguarding procedures were followed, despite the SU’s wishes. Another participant felt that this procedure might be related to the individual being overwhelmed by the number of people attending to the situation: 

*“Because I think like…when you maybe say to someone, oh yeah, we’ve got an ambulance coming,” and then maybe four Police officers and two paramedics…that’s just so overwhelming for them, like…that is a big issue as well, I think.”* (S2P1). 

While participants understood the reasons and importance of safeguarding, several participants discussed whether the police were the right service to call when dealing with someone who is struggling with suicidal thoughts, feelings and behaviours. However, they acknowledged that there were no better options available: 

*“Police aren’t trained to deal with suicidal people. It feels like quite a blunt instrument that we’ve got really, but we don’t…we don’t really have anything better, I suppose. You know, you can’t…you can’t get an ambulance to go out to deal with suicidal people–it would be the Police. Yeah. So, it’s kind of what we have, but, you know, obviously we realise it isn’t ideal.”* (S1P3).

From this account, the participant is clearly reluctant to contact the police as a safeguarding procedure yet are required to in times of suicide risk. This topic was accentuated by frustration by this staff member. Given the level of judgement needed at an individual level, potentially personal opinions could impact safeguarding discussion making. 

#### 3.3.2. Continued Support When SUs Decline Safeguarding

In some situations, participants spoke about how SUs required safeguarding due to imminent risk (typically described as suicide plan within a timeframe) but would not share their personal details with the service or specifically requested that no safeguarding was made. This second example meant that the frontline staff would stay on the call with the user but according to the service ethos were required to respect the personal autonomy of the SU above safeguarding procedure. Although helpline staff are unable to trigger safeguarding responses in such instances, they need to maintain composure and continue supporting the SU in their conversation: 

*“…you talk to somebody for whom you don’t have contact details and you have to almost just, at the end of the session, it’s just there and all you can do is go I gave that person my time and my emotional energy and my support at a point when they needed…”* (S1P1).

Some participants described how they would continue the conversation with SUs, if the risk was not imminent, by exploring with the SU the reasons behind their suicidal thoughts and feelings. This might include offering details of other services from which they may wish to seek help, or encouraging formal help-seeking once the individual feels able to do so. Participants would also remind the SUs that they would be able to call emergency services to help them if they shared their personal details. 

Participants described how being unable to trigger the safeguarding response was challenging for them as professionals. Due to this, sometimes additional support was needed to help manage their own feelings around helplessness.

## 4. Discussion

Qualitative semi-structured interviews with 13 staff across three UK-based helpline organisations identified three key themes associated with identifying, assessing and managing suicide risk among service users (SUs). These were: (i) Starting conversations about suicide; (ii) Identifying and responding to “imminent” suicide risk; and (iii) Responses to suicide risk in relation to safeguarding. Accounts highlighted the importance of creating a safe space for children and young people to openly share feelings and thoughts in relation to self-harm and/or suicide knowing that they will not be judged. By creating this safe space also frontline staff discussed that they felt more able to explore contextual factors that may have led the young person to the point of considering suicide, thus obtaining a holistic picture of a young person’s life. Part of this involved frontline staff identifying the level of suicide risk (or “immediacy” of risk), from which they would judge, based on their own perceptions, how to respond to the SU. In some cases, this would involve evoking safeguarding procedures, such as calling emergency services. Despite different procedures across helplines (e.g., threshold for triggering safeguarding protocol), there were clear commonalities in their practices, such as determining the threshold for imminent suicide risk.

Helplines are among the most commonly accessed source of informal support among young people who seek help for self-harm [30]. Our findings highlight not only the different types and range of services helpline organisations offer to young people but most importantly the distinct role they have in young people’s help-seeking pathway. Their ability to engage with some of the most vulnerable young people is based on the same values that highlight their distinctiveness including accessibility, informality, flexibility and sensitivity to the needs of the community. Building personal capacity, agency and using each interaction to facilitate a supportive, person-centred communication led by the young person’s narrative. Such practices allow the young person to direct the conversation and focus on the thoughts, feelings, and experiences that they perceive as important [20], thus offering a tailored approach. 

A key distinction between helpline services was the protocols surrounding threshold for suicidal behaviour, how this related to safeguarding and relied on frontline staff’s judgement. For example, some frontline staff described how self-harm was not viewed as a specific criterion for suicide risk, while another prioritized the SU’s decision to attempt suicide without intervention over the staff member’s safeguarding judgement. This related to the ethical dilemmas which surround helplines, as different services had different positions and protocols in relation to safeguarding [31]. Previous research has highlighted services, such as Samaritans, who state they are an organization to support emotional distress and prevent suicide, yet hold that callers have the autonomy to make their own decisions regarding suicidal death [31]. This is a similar position to one of the included helplines. Yet, this brings a host of issues, as frontline staff are responsible for making the judgement calls, thereby putting the burden on them. This could result in feelings of guilt, failure, or experiences of vicarious trauma if there is an undesirable end to the contact. This in turn could influence how they respond to other suicidal service users, for example being more likely to breach confidentiality due to safeguarding concerns. Thus, clear guidance and protocols need to be in place as to how each helpline specifically handles suicide risk, and what this means for the frontline staff. 

Helplines have been discussed as an adjunctive or alternative pathway to accessing therapeutic services [22]. In their systematic review, MacDonald et al. [22] highlighted the complexity of mental health pathways for young people, which could impact whether the young person continued or obtained the care needed. One of the unique services offered by helplines was the capacity to offer professional services such as counselling appointments and ongoing, support in addition to their helpline functions, thus enabling a joined-up and tailored response, if and when needed. This tailored and accessible approach–in combination with increased service demand and workforce capacity issues- could provide the context for understanding the high rates of help-seeking among young people for self-harm and suicidal thoughts from the voluntary sector during the pandemic [24]. Indeed, participants in our study highlighted the increased demand for their services during COVID-19, in particular highlighting high levels of anxiety and depression among young people. 

This study has demonstrated that despite no agreed definition of “imminent risk”, a practicing solution has been adopted by frontline staff: “suicide plan within a specific timeframe”. From this, further work can be conducted, firstly to examine if this is a consistent working definition across a larger number of helpline service, secondly, if not where and why are there differences, and finally, how does this relate to safeguarding protocols. Another key direction would be to explore young people’s views and experiences of seeking, accessing and receiving support from helpline organisations during their pathway to care when feeling suicidal, and, if and how these experiences are shaped by the contexts in which they are growing up (e.g., family, peers, schools, local community). 

## 5. Limitations

Recruitment for this study was impacted by the COVID-19 pandemic. Several organisations expressed an interest in taking part, but were unable to due to staff shortages and other challenges they were experiencing. It therefore has to be recognised that our findings may not reflect the practices and policies of other organisations who also work to support children and young people at risk of suicide. 

All interviews were conducted remotely and while this has certain benefits (e.g., flexible scheduling and geographic advantages [32]), it also has a potential to impact on the process of rapport-building between the researcher and the participants, including non-verbal cues being missed [33,34]. Furthermore, it is important to recognize that while these interviews were based on participants’ views and experiences of practice and procedures, direct observations of professional conduct were not made. Thus, it is possible that participant’ accounts and actions have degrees of difference. 

Finally, the lead author was the only one to extensively code all the interviews. Inherently, reflective thematic analysis is subjective; therefore, the final framework is highly related to the lead author’s reflections and perceptions of the data. To reduce the impact of any individual bias, however, all researcher members were included in the thematic framework review and offered their own feedback. 

## 6. Conclusions

This study addressed a critical gap in the literature by exploring the views and experiences of helpline staff’s practice of responding, assessing, and managing suicide risk among service users. This offers insights into how frontline staff of helplines and counselling services perceive their practice and challenges in relation to suicidal presentations. The key message from all participants was ensuring a good relationship and open dialogue which included suicide, this was thought to best engage young people to disclose any risk. While it is important to acknowledge the differences between helpline and counselling services approaches, participants presented a consistent definition used to assess risk and how this related to their responses and management of the service user. Based on these interviews, frontline staff appear well placed to assess and respond to suicide risk, however additional supports are needed to ensure safeguarding which is perceived as acceptable and effective. 

## Figures and Tables

**Table 1 ijerph-19-10887-t001:** Overview of thematic framework and descriptors.

Theme	Subtheme	Descriptor
**Starting conversations about suicide**	Explicitly asking about suicide	Opportunity to explicitly ask about engagement with self-harm or suicide.
	Exploring and understanding	Exploring the context and situation of self-harm and suicide.
**Identifying and responding to “imminent” suicide risk**	Determining immediacy of risk	Unpicking how immediacy of risk is determined. Mechanisms to do this might be through caller description of injury, use of language to determine intention or specifically looking for a suicide plan and the timeliness of this plan.
	Making judgement calls	Frontline staff ultimately have to determine the immediate risk of a caller or client.
**Responses to suicide risk in relation to safeguarding**	Triggering safeguarding	Across all services the safeguarding response was to inform emergency services if the caller or client had shared their details. Critical reflections of calling the police; apprehension.
	Continued support when SUs decline safeguarding	Frontline staff have to stay calm and continue offering their clinical support to suicidal SU.

## Data Availability

The data presented in this study are available on request from the senior author. The data are not publicly available due to funder legal requirements and restrictions.

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
