# Peer review of "Assessing and Responding to Suicide Risk in Children and Young People: Understanding Views and Experiences of Helpline Staff"

_ijerph, 2022, doi:10.3390/ijerph191710887_

Round 1

Reviewer 1 Report

Comments to the Author:

Dear authors,

Thanks for giving me the chance to review your paper, which I found highly relevant to discussions about suicide prevention. It is well-written and clear in its organization. My comments are primarily related to improving the paper in three areas: contextualization of the investigated setting (Introduction), not over-interpreting data (Results), and highlighting the most clear and important findings (Results/Discussion). Below, I detail these comments.

Introduction

In this section, it is sometimes unclear what national/global context you are referring to. Please specify whether you are referring to the situation worldwide or in the UK. (e.g. p. lines 28-36, p. 2, lines 44-52, lines 55-58).

You are right to say that there is limited knowledge on how helpline staff manage suicide risk in young people. However, there is research, which you do not cite, on challenges for helpline staff in general, which I think is a contextualization that would benefit the paper. For example, the work by Pollock et al. (e.g. “Configuring the caller in ambiguous encounters: Volunteer handling of calls to Samaritans emotional support services”) as well as the recent evaluation of Samaritans would be nice to cite to get a sense of the challenges helpline staff is facing. There is also conversation analytic work by Butler and others (e.g. “Advice-implicative Interrogatives: Building “Client-centered” Support in a Children’s Helpline”) and Pudlinski (“Empowerment on warm lines: microanalytical explorations of peer encouragement”) which demonstrates these challenges.

Material & Methods

The description of data collection and analysis is written in a clear way. You could be a little bit more elaborate on how you went from the 4000 codes to a framework. Were the research question formulated before the analysis or as it progressed?

Results

First, I think the result section could be more powerful in presenting the key findings. I think the management of imminent risk (3.2 and 3.3) is a very important finding, and I think you could structure your result section to highlight that more. I think you can cut the first sentence and instead start with saying something about the whole analysis. You could, for instance, start with describing the participants’ descriptions of the institutional (lack of) support in assessing and managing risk. This could be compared to US crisis lines where risk assessment and management is more structured (see Mishara,Resolving Ethical Dilemmas in Suicide Prevention: The Case of Telephone Helpline Rescue Policies). The sentence from p. 6. Lines 221-222: “Overall, the participants presented ‘imminent’ risk as being associated with a suicide plan with a specific timeframe.” Would fit in such an intro to the result section, before you present your themes.

In the analysis (to prepare for the discussion), the implications (positive and negative) of staff being left to judge and manage suicide risk could be commented on. This can be done by highlighting tensions in the data – for example between the excerpt on p. 7, lines 278-282 and lines 286-287. You might also want to bring up that there could potentially be different safeguarding measures, whereas the participants seem to just refer to emergency services.

In the final result section, it is unclear to me if you refer to the same concept of imminent risk as before (suicide plan with a specific time frame).

My second comment about the result section is how you refer to the participants’ accounts. It is important to distinguish between what they do and what they say they do, not least because they sometimes make assessments of what works, which may go against research findings (e.g. asking a lot of questions). Therefore, you need to describe their descriptions as descriptions, not as their practice. Examples can be found on p. 4, line 143, 155, 157-158, p. 5 line 199. By contrast, there are places where you in a really nice way refer to the participants’ account as accounts, such as p. 4, 163-165.

Discussion

Again, it is important to distinguish between practice and interview accounts (e.g. p.8, lines 327-327, 337-339, 342-345). Sometimes it sounds as if you have observed practice or talked to the young people. I also think it is important to discuss the findings around management of high risk, related to the literature I have mentioned about different policies. I think your research has implications here (although they are based on very few interviews of course), but they need to be brought out.

Limitations

You should maybe mention here that you are just studying what they say (in an interview setting) they do and experience.

Minor things

Be consistent in how you present the ‘users’: sometimes clients, sometimes callers, even though you say that you will refer to them as service users. (e.g. Table 1)

Present the data excerpts a bit more. For instance, what is “then” referring to in the first one (p. 4, lines 151-154).

I also wonder if you could pick up more on what the participants say, not just report it or leave it to the reader to analyze (e.g. p. 6 after the first excerpt and after lines 243-247).

Author Response

Dear reviewer

We would like to thank you for the time taken and attention given to reviewing our manuscript. From these comments, we feel the manuscript has really benefited and are very grateful for receiving the feedback.

Based on your comments, we have made the following changes.

Introduction

In this section, it is sometimes unclear what national/global context you are referring to. Please specify whether you are referring to the situation worldwide or in the UK. (e.g. p. lines 28-36, p. 2, lines 44-52, lines 55-58).

We have since clarified on these points, thank you.

You are right to say that there is limited knowledge on how helpline staff manage suicide risk in young people. However, there is research, which you do not cite, on challenges for helpline staff in general, which I think is a contextualization that would benefit the paper. For example, the work by Pollock et al. (e.g. “Configuring the caller in ambiguous encounters: Volunteer handling of calls to Samaritans emotional support services”) as well as the recent evaluation of Samaritans would be nice to cite to get a sense of the challenges helpline staff is facing. There is also conversation analytic work by Butler and others (e.g. “Advice-implicative Interrogatives: Building “Client-centered” Support in a Children’s Helpline”) and Pudlinski (“Empowerment on warm lines: microanalytical explorations of peer encouragement”) which demonstrates these challenges.

Thank you for your comments. Wehave enhanced our introduction to more clearly contextualise challenges facing helpline staff.

Despite being well-placed for user access, helplines face their own challenges for staff and volunteers. For example, helpline staff need to determine the level of emotional need from callers [23] which requires a level of rapid assessment of genuine dis-tress while also dealing with potentially abusive or inappropriate callers [23]. Furthermore, volunteers often need to make judgements of who is in need of assistance which is directly against the helpline ethos of being non-judgment [23], this may then damage supportive relationships with callers. It seems that the limited face-to-face dynamic of helplines, although helpful for preserving anonymity, could create barriers for staff or volunteers to gain a clear representation of distress and callers feeling as though they are being dis-missed [23]. During COVID-19 pandemic, additional demand impacted helplines. Between 23 March and 10 May 2020, the NSPCC Childline service delivered 30,868 counselling sessions to children and young people, 36% of which were about mental and emotional health concerns, 13% were about suicidal thoughts and 6% about self-harm [24]. Samaritans also saw a 12% increase in calls made between 2am-6am compared with previous years and longer phone sessions (40%) [25]. This increased demand for helpline services means that staff and volunteers are under greater strain to aid a higher number of callers while still making precise and calculated evaluation of caller wellbeing.

Material & Methods

The description of data collection and analysis is written in a clear way. You could be a little bit more elaborate on how you went from the 4000 codes to a framework. Were the research question formulated before the analysis or as it progressed?

This has now been clarified;

Codes were clustered to develop a meaningful preliminary framework in line with the re-search questions, which were formulated prior to engaging with the data.

Results

First, I think the result section could be more powerful in presenting the key findings. I think the management of imminent risk (3.2 and 3.3) is a very important finding, and I think you could structure your result section to highlight that more. I think you can cut the first sentence and instead start with saying something about the whole analysis. You could, for instance, start with describing the participants’ descriptions of the institutional (lack of) support in assessing and managing risk. This could be compared to US crisis lines where risk assessment and management is more structured (see Mishara, “Resolving Ethical Dilemmas in Suicide Prevention: The Case of Telephone Helpline Rescue Policies).

The sentence from p. 6. Lines 221-222: “Overall, the participants presented ‘imminent’ risk as being associated with a suicide plan with a specific timeframe.” Would fit in such an intro to the result section, before you present your themes.

Thank you for this comment. We have now made this change to the results;

Across the services, there was no firm consensus of what qualified “imminent” or “immediate” risk; therefore, participants described their own definitions. This was typically a suicide plan within a specific timeframe. Identifying and responding to risk was clearly related to the varying approaches to suicide risk and autonomy taken by the ser-vices, e.g. reactive safeguarding. It was obvious that there was not one single way universe-sally to identify or respond to “imminent” risk.

However, we have chosen to add Mishara’s paper in the discussion for consistency of the results section.

In the analysis (to prepare for the discussion), the implications (positive and negative) of staff being left to judge and manage suicide risk could be commented on. This can be done by highlighting tensions in the data – for example between the excerpt on p. 7, lines 278-282 and lines 286-287. You might also want to bring up that there could potentially be different safeguarding measures, whereas the participants seem to just refer to emergency services.

Thank you for this comment. We have made minor changes to the results but have commented on this within the discussion as well.

Participants discussed the importance of maintaining the relationship with the SU, which was often benefitted by the frontline staff member being able to judge the context and situ-ation themselves, rather than follow a closed-script. This meant that they were able to solidify relationships and be transpar-ent when they felt there was a need to break confidentiality:

….

Despite clear communication and policies regarding safeguarding procedures, par-ticipants were concerned about the impact of calling emergency services for SUs, and con-tributing to their distress further. This was a challenge for some participants as they weighed the judgment of risk compared to further harm to the SU. Sometimes SUs would request that police were not called:

In the final result section, it is unclear to me if you refer to the same concept of imminent risk as before (suicide plan with a specific time frame).

Thank you for this comment. We have since clarified.

In some situations, participants spoke about how SUs required safeguarding due to imminent risk (typically described as suicide plan within a timeframe) but would not share their personal details with the service.

My second comment about the result section is how you refer to the participants’ accounts. It is important to distinguish between what they do and what they say they do, not least because they sometimes make assessments of what works, which may go against research findings (e.g. asking a lot of questions). Therefore, you need to describe their descriptions as descriptions, not as their practice. Examples can be found on p. 4, line 143, 155, 157-158, p. 5 line 199. By contrast, there are places where you in a really nice way refer to the participants’ account as accounts, such as p. 4, 163-165.

Thank you for this comment. We have since gone through the results section again to highlight this difference. We have also included this distinction in the limitations.

Discussion

Again, it is important to distinguish between practice and interview accounts (e.g. p.8, lines 327-327, 337-339, 342-345). Sometimes it sounds as if you have observed practice or talked to the young people. I also think it is important to discuss the findings around management of high risk, related to the literature I have mentioned about different policies. I think your research has implications here (although they are based on very few interviews of course), but they need to be brought out.

Thank you for your comments.

We have clarified in the first paragraph that the results are based on accounts rather than directly knowing what frontline staff do through observation.

A section has been included about ethical dilemmas at helplines and the impact to staff;

A key distinction between helpline services was the protocols surrounding threshold for suicidal behaviour, how this related to safeguarding and relied on frontline staff’s judgement. For example, some frontline staff described how self-harm was not viewed as a specific criterion for suicide risk, while another prioritized the SU’s decision to attempt suicide without intervention over the staff member’s safeguarding judgement. This related to the ethical dilemmas which surround helplines, as different services had different positions and protocols in relation to safeguarding [31]. Previous research has highlighted services, such as Samaritans, who state they are an organization to support emotional distress and prevent suicide, yet hold that callers have the autonomy to make their own decisions regarding suicidal death [31]. This is a similar position to one of the included helplines. Yet this brings a host of issues, as frontline staff are responsible for making the judgement calls, thereby putting the burden on them. This could result in feelings of guilt, failure, or experiences of vicarious trauma if there is an undesirable end to the contact. This in turn could influence how they respond to other suicidal service users, for example being more likely to breach confidentiality due to safeguarding concerns. Thus, clear guidance and protocols need to be in place as to how each helpline specifically handles suicide risk, and what this means for the frontline staff.   

We have included a brief statement about risk-management research implications;

This study has demonstrated that despite no agreed definition of “imminent risk”, a practicing solution has been adopted by frontline staff: “suicide plan within a specific timeframe”. From this, further work can be conducted, firstly to examine if this is a consistent working definition across a larger number of helpline service, secondly, if not where and why are there differences, and finally, how does this relate to safeguarding protocols.

Limitations

You should maybe mention here that you are just studying what they say (in an interview setting) they do and experience.

Thank you for this comment. This has now been included.

Furthermore, it is important to recognize that while these interviews were based on participants’ views and experiences of practice and procedures, direct observations of professional conduct were not made

Minor things

Be consistent in how you present the ‘users’: sometimes clients, sometimes callers, even though you say that you will refer to them as service users. (e.g. Table 1)

Thank you – this has now been changed

Present the data excerpts a bit more. For instance, what is “then” referring to in the first one (p. 4, lines 151-154).

Thank you – this has now been changed

I also wonder if you could pick up more on what the participants say, not just report it or leave it to the reader to analyze (e.g. p. 6 after the first excerpt and after lines 243-247).

Thank you – this has now been changed

Reviewer 2 Report

Assessing and responding to suicide risk in children and young people: Understanding views and experiences of help-line staff

Thank you for the opportunity to review this paper, which examined the views and experiences of helpline organizations in identifying, assessing, and managing suicide risk among children and young people. Findings suggested 3 themes, including starting conversations about suicide, identifying and responding to imminent suicide risk, and responses to suicide risk in relation to safeguarding. I thought this manuscript was well-written. This study has important implications for understanding the important role of helpline organizations and how they can be better structured to help young people at-risk for suicide. I have a few minor recommendations for improving the manuscript.

Introduction

1.       In the second paragraph, when discussing some of the barriers to accessing mental health services among young people, can the authors describe the role of parents? For instance, are young people required to obtain parental consent, are parents notified, etc. when young people seek these services?

2.       In the third paragraph, it is stated that young people are more willing and likely to access support from community-based services. Could the authors be more specific here? More willing/likely than whom or what?

3.       The introduction would be strengthened by noting the potential implications for the aims – ie, why do we need to know challenges faced among helpline staff?

Methods

1.       The fact that only one person coded all of the interviews is a limitation that should be addressed. Was a second coder ever included for reliability?

Discussion

1.       As noted in the methods, please include the use of one coder as a limitation.

Author Response

Dear reviewer

We would like to thank you for the time taken and attention given to reviewing our manuscript. From these comments, we feel the manuscript has really benefited and are very grateful for receiving the feedback.

Based on your comments, we have made the following changes.

  1. In the second paragraph, when discussing some of the barriers to accessing mental health services among young people, can the authors describe the role of parents? For instance, are young people required to obtain parental consent, are parents notified, etc. when young people seek these services?

Thank you for this comment, we have now included examples of parental barriers.

Disclosure may be a significant challenge among children and adolescents, due to fears that parents or guardians will be informed. For example, in some areas of the U.S. parents have legal access to adolescents’ electronic health records until they are 18 years old or are notified when services are sought [15], thereby nullifying the individual’s confidentiality when seeking professional help.  

  1. In the third paragraph, it is stated that young people are more willing and likely to access support from community-based services. Could the authors be more specific here? More willing/likely than whom or what?

Thank you for this comment, we have included more specificity in this section.

Young people are more willing and likely to access support from community-based services (e.g. online forums, peer support groups, helplines) when feeling suicidal [8, 14]. From a previous meta-analysis, when young people did seek help, it was more often from community-based supports (40-68%) rather than medical or mental health services, which were classed as professional help-seeking (below 50%) [8].

  1. The introduction would be strengthened by noting the potential implications for the aims – ie, why do we need to know challenges faced among helpline staff?

Thank you for this comment, we have enhanced this in the introduction;

Despite being well-placed for user access, helplines face their own challenges for staff and volunteers. For example, helpline staff need to determine the level of emotional need from callers [23] which requires a level of rapid assessment of genuine dis-tress while also dealing with potentially abusive or inappropriate callers [23]. Furthermore, volunteers often need to make judgements of who is in need of assistance which is directly against the helpline ethos of being non-judgment [23], this may then damage supportive relationships with callers. It seems that the limited face-to-face dy-namic of helplines, although helpful for preserving anonymity, could create barriers for staff or volunteers to gain a clear representation of distress and callers feeling as though they are being dis-missed [23]. During COVID-19 pandemic, additional demand impacted helplines. Between 23 March and 10 May 2020, the NSPCC Childline service delivered 30,868 counselling sessions to children and young people, 36% of which were about mental and emotional health concerns, 13% were about suicidal thoughts and 6% about self-harm [24]. Samaritans also saw a 12% increase in calls made between 2am-6am compared with previous years and longer phone sessions (40%) [25]. This increased demand for helpline services means that staff and volunteers are under greater strain to aid a higher number of callers while still making precise and calculated evaluation of caller wellbeing.

Helplines are a common source of help-seeking among suicidal young people. Given the recent heightened demand and challenges for helpline services, it is important to understand how helpline staff and volunteers identify, assess, and manage suicide risk among young people; .  as well as the challenges staff face when responding to suicidal young people, and how they manage these within an organisational structure. Having a better under-standing of the current practice and challenges facing these services, could help us identify examples of good practice, examples where practice could be optimised; thus, highlighting opportunities for improvement through, for example, the provision of tailored training and/or resources. The aim of this study was to explore the views and experiences of helpline staff and volunteers when identifying, assessing, and managing suicide risk among young people in order to better understand their current practice.

Methods

  1. The fact that only one person coded all of the interviews is a limitation that should be addressed. Was a second coder ever included for reliability?

Thank you for highlighting this. More information has been included in this section, however only one coder was used – the second author reviewed these codes and the framework.

Following transcription, all transcripts were imported into NVivo 12, and the first author began coding these inductively. Extensive coding of the transcripts was performed, generating over 4000 individual codes. Codes were clustered to develop a meaningful preliminary framework in line with the research questions. The codes and preliminary framework were then reviewed with the second author.

Discussion

  1. As noted in the methods, please include the use of one coder as a limitation.

This has been added as a limitation.

Finally, the lead author was the only one to extensively code all the interviews. Inherently, reflective thematic analysis is subjective, therefore the final framework is highly related to the lead author’s reflections and perceptions of the data. To reduce the impact of any individual bias however, all researcher members were included in the thematic framework review and offered their own feedback.

Reviewer 3 Report

Thank you for an opportunity to comment on the mss on assessing and responding to suicide risk in children and young people: Understanding views and experiences of helpline staff. The text is well written and presents useful information. I have a few comments/queries:

1. 1st paragraph in the Introduction: Can the authors clarify whether the data presented refer specifically to the UK?

2. The Introduction includes many references to the Covid-19 pandemic. How doe they relate to the study aims and methods?

3. Please, provide information on the eligibility criteria for the study participants. 

Author Response

Dear reviewer

We would like to thank you for the time taken and attention given to reviewing our manuscript. From these comments, we feel the manuscript has really benefited and are very grateful for receiving the feedback.

Based on your comments, we have made the following changes.

1st paragraph in the Introduction: Can the authors clarify whether the data presented refer specifically to the UK?

Thank you for this comment, we have now clearly stated the locations of the citations.

The Introduction includes many references to the Covid-19 pandemic. How do they relate to the study aims and methods?

We have included more information to streamline this argument.

The study took place during the COVID-19 pandemic. Helpline professionals report-ed the increasing demand of children and young people seeking help from their organisations during this period. Furthermore, the distress reported by young people appears to be on the rise, with more complex needs being discussed [26]. In addition, services struggled with staff absences, and changing working practice. This is in line with challenges experienced across mental health services in the UK during this time. Due to this increased demand of both complex needs and general help-seeking, it seems timely to have a deeper insight into how helplines deal with young people seeking help for suicide. Due to the pandemic, all methodology took place online rather than in-person settings.

  1. Please, provide information on the eligibility criteria for the study participants. 

We have now included this criteria.

The eligibility criteria for participants was; i) to currently be working at a helpline or online counselling service, ii) to have experience dealing with service users with suicidal experiences, and iii) to be able to take part in an English-speaking online interview.

Round 2

Reviewer 1 Report

Thank you for the revision and your explanations. I just have a few minor comments, which you can see in my comments in the attached document.

Author Response

Thank you for your comments. We appreciate the time taken to review and submit your feedback. In line with this, we have revised the manuscript language as suggested and enhanced the line of description in the results to meet your comment; 

"here I think you shoudl be clearer that they think this benefits the relationship to the SU."

"taff felt that this enabled SUs to have a greater sense of trust in the discussion, enhancing rapport and satisfaction with the contact, as well as allowing for open disclosure of suicide risk. According to the staff, this flexibility allowed them to solidify relationships..."